# Constraints on Enhanced Weathering and related carbon sequestration – a cropland mesocosm approach

[1]Thorben Amann, [1]Jens Hartmann, [2]Eric Struyf, [1]Wagner de Oliveira Garcia, [3]Elke K. Fischer, [4]Ivan Janssens, [2]Patrick Meire, [2]Jonas Schoelynck

[1]Institute for Geology, Center for Earth System Research and Sustainability, University of Hamburg, Germany
[2]University of Antwerp, Department of Biology, Ecosystem Management Research Group, Universiteitsplein 1C, B-2610 Wilrijk, Belgium
[3]Institute of Geography, Center for Earth System Research and Sustainability, University of Hamburg, Germany
[4]University of Antwerp, Research Centre of Excellence Global Change Ecology, Universiteitsplein 1, B-2610 Wilrijk, Belgium

*Correspondence to*: Thorben Amann (science@thorbenamann.de)

**Abstract.** The weathering of silicates is a major control on atmospheric $CO_2$ at geologic time scales. It was proposed to enhance this process to actively remove $CO_2$ from the atmosphere. While there are some studies that propose and theoretically analyze the application of rock powder on agricultural land, results from field experiments are still scarce.

In order to evaluate the efficiency and side effects of Enhanced Weathering, a mesocosm experiment was set up and agricultural soil from Belgium was amended with olivine-bearing dunite ground to two different grain sizes, while distinguishing setups with and without crops.

Based on measurements of Mg, Si, pH, and DIC, the additional weathering effect of olivine could be confirmed. Calculated weathering rates are up to three orders of magnitude lower than found in other studies. The calculated $CO_2$ consumption by weathering based on the outlet water of the mesocosm systems was low with 2.3 - 4.9 $CO_2$ t $km^{-2}$ $a^{-1}$ if compared with previous theoretical estimates. Suspected causes were the removal or dilution of Mg as a weathering product by processes like adsorption, mineralization, plant uptake, evapotranspiration, and/or preferential flow, not specifically addressed in previous Enhanced Weathering experiments for $CO_2$ consumption. The observation that Mg concentrations in the upper soil layers were a about magnitude higher than in the outlet water indicates that a careful tracking of weathering indicators like Mg in the field is essential for a precise estimate of the $CO_2$ consumption potential of Enhanced Weathering, specifically under global deployment scenarios with a high diversity of ecosystem settings. Porewater Mg/Si molar ratios suggest that dissolved Si is re-precipitating, forming a cation depleted Si layer on the reactive mineral surface of freshly ground rocks.

The release of potentially harmful trace elements is an acknowledged side effect of Enhanced Weathering. Primarily Ni and Cr are elevated in the soil solution, while Ni concentrations exceed the limits of drinking water quality. The use of olivine, rich in Ni and Cr, is not recommend and alternative rock sources are suggested for the application.

# 1 Introduction

The application of rock powder on agricultural soils has long been used to improve soil properties to achieve a productivity increase (De Villiers, 1961; Kronberg, 1977; Leonardos et al., 1987; Anda et al., 2015a, 2013; Anda et al., 2015b; Shamshuddin and Anda, 2012), predominantly in the form of liming. The application of carbonate rock powder to agricultural soils is a process to adjust soil pH (Cregan et al., 1989) in order to increase crop production (Haynes and Naidu, 1998). Besides pH adjustment, the additional release of cations and anions into the soil-rock system alters the chemical composition of the soil solution. Alternative amendment materials are gaining increased attention, one of which is silicate rock powder. Silicate rocks can provide geogenic nutrients via the chemical weathering of the additional minerals (Hartmann et al., 2013; Van Straaten, 2006). On top of that, it has the potential additional advantage of enhancing atmospheric $CO_2$ sequestration: on geological time scales, natural silicate weathering is one of the most important controls on atmospheric C concentrations (Berner, 2003). The silicate weathering process releases cations like $Mg^{2+}$, $Ca^{2+}$ and others, and $CO_2$ is stored as alkalinity in the ocean, whereas carbonate weathering yields no net $CO_2$ uptake on longer timescales (Hartmann et al., 2013). Enhanced silicate weathering has therefore been put forward as a method/technique with strong potential to contribute to climate mitigation. In order to achieve COP21 atmospheric $CO_2$ concentration targets, it becomes more likely that not only emission reduction is required (Fuss et al., 2014; Rogelj et al., 2018; Peters, 2016). Focus should also be put on applying effective $CO_2$ sequestration techniques (Sanderson et al., 2016; Beerling et al., 2018; Minx et al., 2018).

With dwindling resources of rocks with concentrated content of widely applied macro-nutrients, which might lead to a shortage of traditional fertilizers (Cordell et al., 2009; Manning, 2015), geogenic nutrient replacement by Enhanced Weathering will become a valid alternative to supply not only phosphorus or potassium but also further geogenic nutrients, with potentially important local impact on food security (van Straaten, 2002; Cordell et al., 2009). In addition, alternative regional fertilizer concepts for certain regions need to be developed to enhance productivity, as for example in Africa (Ciceri and Allanore, 2019).

However, the application of silicate rock products requires knowledge of soil mineral properties, hydrology, soil solution composition, and element uptake by plants to enable predictions on its consequences. Specifically this knowledge is lacking at the broader scheme (Beerling et al., 2018; Beerling, 2017; Kantola et al., 2017; Edwards et al., 2017; Taylor et al., 2017), despite several experiments in the past (Anda et al., 2015a, 2013; Shamshuddin and Anda, 2012; Shamshuddin et al., 2011). One of the main gaps is the evolution of soil solution composition and its migration in the treated soil, considering a broad variety of possible combinations of soil type, rock product, and plant species (Hartmann et al., 2013). The time scale at which changes in weathering fluxes can be expected at the scale of large catchments was shown for the Mississippi, where alkalinity fluxes increased by more than 50% over less than a century, which was partly attributed to liming and land management processes (Raymond and Cole, 2003; Raymond and Hamilton, 2018). In general, past land use change and management of catchments can affect the chemical baselines of rivers draining to the ocean over decades (Hartmann et al., 2011; Hartmann et

al., 2007; Meybeck, 2003; Radach and Pätsch, 2007). The large-scale application of rock products will likely lead to an alteration of river chemistry, and consequences for adjacent coastal zones remain to be assessed.

In the future, increasing food and bioenergy demand will probably lead to more efforts to improve soil conditions for optimized biomass production (Fuss et al., 2018; IPCC, 2019). The future application of customized rock products to provide slow release

geogenic nutrient fertilizers, adjust pH, increase cation exchange capacities (CEC), or adjust soil hydrology is therefore likely. The replenishment of geogenic macro- and micronutrients is needed because the natural supply cannot keep up with the permanent removal from the soil-rock system under intensive harvest scenarios for crops or timber (de Oliveira Garcia et al., 2018; Van Straaten, 2006). The application of rock products will therefore change the fluxes of elements within and from the soils, while being mediated by the biological pump.

One of the key issues is the dissolution rate of applied rock material. While the kinetics are relatively well understood at the laboratory scale for singular minerals (Rosso and Rimstidt, 2000; Wogelius and Walther, 1992), the dissolution rate of a rock powder mixture as soil amendment, with fresh surfaces, which have not been in contact with an aquatic phase before, is nearly unknown. Several points of the rock powder application on soils have to be considered. First, the upper parts of soils are not permanently saturated with water, which may lead to mineral dissolution-precipitation reactions. Second, it can be expected

that mineral surfaces initially need to equilibrate with the new system and varying water content, and that dissolution rates of minerals will be different from those being in long-term equilibrium within the natural soil system. Third, trace elements from the applied rock material will eventually be released and migrate downwards, re-precipitated if oversaturation with a specific mineral phase occurs, or adsorbed to soil minerals or organic matter.

To understand these processes in an agricultural setting with typical crops, dunite could serve as model rock material,

containing often more than 90% of olivine, a mineral often used as model mineral to theoretically study effects of Enhanced Weathering (Schuiling and Krijgsman, 2006; Hartmann et al., 2013; Köhler et al., 2010; Taylor et al., 2015; Renforth et al., 2015; Montserrat et al., 2017). Using near-monomineralic rocks decreases the complexity of observable effects. Discussed alternatives like basalt, have a much greater complexity (considerable quantities of plagioclase and pyroxene, and to a lesser extent olivine and other trace minerals). In addition, basalt has the potential to provide the nutrient phosphorus, which is

typically low in dunite. The release of phosphorus could potentially influence plant-weathering interactions in the soil, complicating the analysis of the weathering process. In our present study, we applied a dunite to agricultural soils to quantify the impact on inorganic carbon and dissolved silica fluxes in the presence and absence of crop plants.

We studied the release of the major elements Mg and Si predominantly derived from Mg-olivine, as indicators for the inorganic $CO_2$ sequestration potential, and assessed whether the release of elements into the soil solution occurs stoichiometrically, or

whether a secondary layer covering the fresh surfaces of minerals will develop, potentially enriched in Si and depleted in Mg (Daval et al., 2013a; Hellmann et al., 2012; Pokrovsky and Schott, 2000), which could influence weathering and subsequently sequestration rates. In addition, the release of trace metals was used to understand how these behave in a near natural environment to evaluate the impacts on the environment.

## 2 Methods

### 2.1 Mesocosm setup

In October 2013, a fully-replicated setup (five replicates per treatment combination) of mesocosms was built up and left running for 730 days. The experimental setup was not specifically tailored to this study of weathering fluxes as we piggybacked
on an experiment to evaluate elemental cycling into plants. Here we report on data of the first year. Rain barrel type mesocosms with a diameter of 46 cm were filled with a natural loamy sandy soil from Belgium (detailed characterisation including grain size distribution in Suppl. Mat. S1). Controlled factors were the application of olivine-rich dunite (henceforth referred to as olivine amendment) in the top layer of the soils (22 kg m$^{-2}$, a high mass to induce observable effects, and a similar value as the maximum mass applied in an experiment by ten Berge et al. (2012)) using two different olivine grain size fractions (roughly
coarse sand and fine sand to silt), two crop plants (wheat and barley) and two irrigation regimes (daily and weekly precipitation), while the total amount of rain was equal (about 800 mm a$^{-1}$). Controls were established by using the same setup without olivine application (blanks) and without plants. Waters were sampled at 1.5, 12.5, and 24.5 cm depth and at the bottom of the mesocosms (Fig. 1).

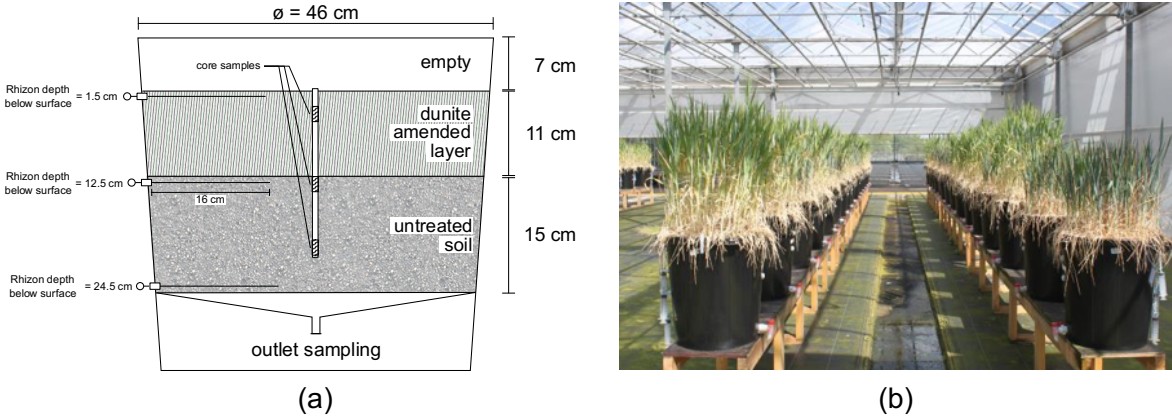

(a)    (b)

**Fig. 1: (a) Schematic mesocosm configuration; (b) status of the experiment in April 2014 (6 months in).**

### 2.2 Material

The experiment material was produced from dunite rock, containing approximately 90% olivine, of which 92% are forsterite (Mg endmember olivine). The rest is comprised of lizardite (Mg-rich serpentine), Cr containing chlorite (including chromite or chrome-spinell inclusions) and traces of chabazite (zeolite group) and Mg-hornblende (amphibole), determined by energy dispersive X-ray spectrometry (Zeiss LEO 1455 VP coupled with an EDX detector by Oxford Instruments). It originates from the Almklovdalen peridotite complex (Åheim mineral deposit mined by North Cape Minerals Company, Norway). More
insights on the geochemistry of the material can be found in Hövelmann et al. (2012), and Beyer (2006) describes the geological setting. The bulk chemical composition (Panalytical Magix Pro wavelength dispersive X-ray fluorescence (XRF) analysis) of the sample is given in Tab. 1. The particle size distribution of the two grainsize classes used were analysed by Sympatec Helos

KFMagic laser granulometry (Tab. 2). The sample was analyzed for specific surface area, measured using $N_2$ and Kr adsorption during BET analyses (Brunauer et al., 1938) with a Quantachrome autosorb iQ (Tab. 2). Only the Kr based measurements were used in calculations since the use of Kr ensures more precise results especially at lower surface areas (Naderi, 2015).

5    **Tab. 1: Geochemical composition of source dunite, derived from XRF runs (n=3). *Total iron ($Fe^{2+}$ + $Fe^{3+}$) is reported as $Fe_2O_3$.**

| Oxide | mass-% | s.d. | | Element | mass [mg kg$^{-1}$] | s.d. |
|---|---|---|---|---|---|---|
| $SiO_2$ | 40.14 | 0.05 | | Ba | 55 | 9 |
| $Al_2O_3$ | 0.70 | 0.01 | | Ce | 8 | 1 |
| $Fe_2O_3$* | 6.75 | 0.01 | | Co | 105 | 2 |
| MnO | 0.09 | 0.00 | | Cr | 2156 | 3 |
| MgO | 44.99 | 0.25 | | Cu | 13 | 2 |
| CaO | 0.40 | 0.00 | | Ga | 3 | 1 |
| $Na_2O$ | 0.03 | 0.00 | | La | 0 | 0 |
| $K_2O$ | 0.06 | 0.00 | | Nb | 1 | 1 |
| $TiO_2$ | 0.01 | 0.00 | | Nd | 29 | 5 |
| $P_2O_5$ | 0.01 | 0.00 | | Ni | 2889 | 18 |
| $SO_3$ | 0.00 | 0.00 | | Pb | 1 | 1 |
| LOI | 6.48 | 0.00 | | Rb | 4 | 2 |
| total | 99.68 | | | Sc | 9 | 1 |
| | | | | Sr | 56 | 1 |
| | | | | Th | 1 | 2 |
| | | | | U | 5 | 4 |
| | | | | V | 31 | 3 |
| | | | | Y | 3 | 1 |
| | | | | Zn | 36 | 2 |
| | | | | Zr | 0 | 0 |

**Tab. 2: Specific surface area of the source material, derived by BET analyses, as well as grain size distribution characteristics. [a]sieve mesh at which 20% are retained, thus 80% being smaller than the given diameter; [b]class with the largest class weight; [c]this class is divided into five smaller classes but was summed to show the share below 1 μm.**

| grain size category | Specific surface area, N based [m$^2$ g$^{-1}$] | Specific surface area, Kr based [m$^2$ g$^{-1}$] | p80[a] [μm] | dominating class[b] [μm] | Share of dominating class [%] | Smallest class [μm] | Share of smallest class [%] |
|---|---|---|---|---|---|---|---|
| fine | 9.53±0.43 | 14.75±0.24 | 43.5 | 25.5 | 6.7±0.02 | < 0.9[c] | 2.66±1.51 |
| coarse | 1.06±0.10 | 1.61±0.03 | 1020 | 720 | 10.8±0.7 | < 18 | 1.51±0.03 |

10    **2.3 Analysis**

The sampled pore and outlet waters were filtered through 0.45 μm nitrocellulose Chromafil syringe filters (A-45/25) into sample bottles and stored cool (4°C) until analysis. Soil material was obtained by extracting sediment cores (20 cm long and 28 mm in diameter) using a hammer auger with a removable plastic lining (Eijkelkamp 04.15.SA Foil sampler, Giesbeek, The Netherlands). In each container, one core was taken at the centre. Immediately after sampling, each core was sub-sectioned

15    into ten slices of 2 cm, packed in vacuum plastic bags and stored cool (4°C) on return to the laboratory. Sediment samples were dried for 72 h at 40°C and homogenized by manual grinding. Three slices were analysed: slice 2 (-2 to -4 cm; in dunite

amended layer); slice 6 (-10 to -12 cm; just below dunite amended layer) and slice 10 (-18 to -20 cm; in untreated soil below dunite amended layer).

### 2.3.1 Silica and magnesium

Dissolved silica (Si) and magnesium (Mg) were measured with an inductively coupled plasma atomic emission spectrophotometer (ICP-AES, Thermo Scientific, ICAP 6000 Series).

### 2.3.2 pH

Water from the bottom outlet was drained into a bucket and pH was measured using a WTW pH meter, calibrated with three NIST buffer standards (pH 4, 7, and 10).

### 2.3.3 Trace metals

In the soil solution as well as in the soil material, concentrations of aluminium (Al), barium (Ba) chromium (Cr), cobalt (Co), iron (Fe), manganese (Mn), nickel (Ni), strontium (Sr), and zinc (Zn) were analysed using inductively coupled plasma atomic emission spectroscopy (ICP-AES, Optima 2100, Perkin Elmer).

For analysis of the total content of substances within the soil material, a digestive procedure was done according to Heinrichs and Herrmann (2013). In brief, soil was dried at 40°C and milled with a planetary ball mill. 150 mg of soil was weighed into PTFE crucibles and a mixture of 4 mL nitric acid (65 % *Suprapur* grade), 2 mL hydrofluoric acid (40 % *Suprapur* grade) and 2 mL perchloric acid (70 % *Suprapur* grade) was added. The crucibles were sealed and placed for 10 h in a closed digestion aperture (*PICOTRACE*) at 170°C to ensure complete dissolution. Subsequently, the acids were vaporised in a closed system and the residues were dissolved with 2 mL nitric acid (65 % *Suprapur* grade), 0.6 mL hydrochloric acid (37 % *Suprapur* grade) and 20 mL high-purity water at 90°C for 1 h. The solutions were standardised to 50 mL with high-purity water and underwent atomic emission spectroscopy (ICP-AES) as described above.

### 2.3.4 Dissolved inorganic carbon (DIC)

DIC was measured with a Picarro G2131-i cavity ring-down spectrometer coupled to a preparation device (AutoMate FX, Inc.) for discrete sample measurement. Not at all sampling times, enough sample volume was available for DIC analyses, as priority was given to other major compounds based on the original purpose of the experiment. Samples were preserved with $HgCl_2$ and stored dark and cool until analysis.

### 2.4 Calculation of weathering and CO₂ sequestration rate

The average flux of Mg from dunite amended soils at the outlet can be calculated by

$$Flux_{Mg^{2+}} = ([Mg^{2+}]_{treated} - [Mg^{2+}]_{untreated}) \times q \qquad \text{Eq. 1}$$

With $q$ as water volume discharged at the outlet per sampled time interval.

The sequestration rate can subsequently be calculated by

$$CO_2 \ sequestration = \frac{Flux_{Mg^{2+}} \times molweight_{Mg}}{fraction \ of \ Mg \ in \ olivine} \times RCO_2 \times \omega \qquad \text{Eq. 2}$$

With a fraction of Mg in olivine of about 1.8 (inferred from XRF analysis, Tab. 1). $RCO_2$ is the theoretical maximum uptake of $CO_2$ in tons per ton of rock (1.25), which is corrected by $\omega$ (= 0.85), to account for carbonate system equilibration in the ocean (after Renforth, 2012; Renforth et al., 2013 and references therein). The global $CO_2$ sequestration potential was then calculated by multiplying with the available arable land in an optimistic and a pessimistic scenario (Moosdorf et al., 2014).

The weathering rate can be estimated by

$$Weathering \ rate \ R \left[ \frac{mol \ Olivine}{m^2 s} \right] = \frac{\frac{Flux_{Mg^{2+}}}{1.8}}{applied \ mass_{olivine} \ \times \ specific \ surface \ area \times t} \qquad \text{Eq. 3}$$

The numerator converts the molar flux of Mg to molar flux of olivine (1.8 mol Mg per 1 mol olivine). Time factor $t$ is used to convert the flux measured in 340 days to seconds.

## 2.5 Calculation of the amorphous Si layer

The Mg depleted and Si enriched layer that forms during the dissolution process (Daval et al., 2011) was roughly estimated using the release of Mg in conjunction with the Mg/Si ratio and the available surface area of the forsterite:

The mass of $SiO_2$ that precipitated per m² and year as amorphous Si can be estimated by

$$m_{SiO_{2}amorph.} \left[ \frac{g \ SiO_2}{m^2 a} \right] = \left( \frac{R_{Mg}}{Mg/Si_{theoretical}} - R_{Si} \right) \times M_{SiO_2} \times t \qquad \text{Eq. 4}$$

with the dissolution (weathering) rates $R_{Mg}$, $R_{Si}$, calculated from experimental data, the theoretical Mg/Si ratio (1.8), $M_{SiO2}$ as the molar mass of $SiO_2$, and time factor $t$ to convert seconds to years.

The depletion layer thickness can then be calculated as

$$growth \ rate \ of \ SiO_2 \ layer \left[ \frac{nm}{a} \right] = \frac{m_{SiO_{2}amorph.}}{\rho_{SiO_{2}amorph.} \times \left( 1 - \varphi_{SiO_{2}amorph.} \right)} \times 10^9 \qquad \text{Eq. 5}$$

with the density $\rho_{SiO2amorph}$ as 2.23 x $10^6$ g m$^{-3}$ (Iler, 1979) with the porosity $\varphi_{SiO2amorph}$ as 0.3 (20-40 %; Maher et al., 2016).

## 3 Results

### 3.1 Hydrology

Two rain regimes, with daily and with weekly rainfall, delivering the same total annual precipitation volume, were used. Since there were no significant and/or systematic differences between results of both rain treatments (Suppl. Fig. S2-1), all discussed data integrate values from both rain treatments. After the experiment start, it took between 7 and 23 days until water reached

the bottom of the mesocosms. The amount of irrigation water and the water collected at the outlet of each barrel were used to roughly estimating the loss of water through evaporation and transpiration, not accounting for water storage in biomass and changes in soil water storage capacity. Sample volume which could be extracted varied. Between days 200 and 300, growth of plants and elevated ambient temperatures caused strong evapotranspiration, which reduce the outflowing water volume to

a minimum. At these times, no or only a little sample volume could be obtained. Data clearly shows elevated evapotranspiration in the mesocosm seeded with crops (Fig. 2).

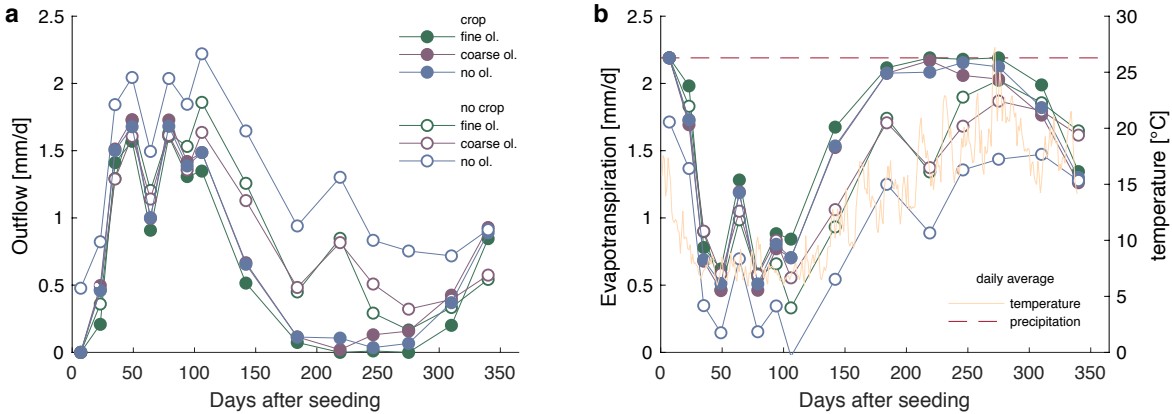

**Fig. 2: a) Water flow from outlet, values refer to daily fluxes from the preceding interval. b) Average daily temperature in the greenhouse and evapotranspiration, calculated from the difference of precipitation input and barrel outflow, relative to precipitation.**

### 3.2 Release patterns of weathering tracers

The fine olivine fraction shows about a ninefold higher specific surface area in both krypton- and nitrogen-based measurements than the coarse fraction (Tab. 2). Observed results are therefore differentiated by fine and coarse rock treatment. Elevated

concentrations of the major studied parameters DIC, Mg, and Si are only observed at 1.5 and 12.5 cm depth, with the largest increase in the top sampling point if compared to the base level setup without olivine. Changes in DIC and Mg concentrations are most pronounced in the mesocosms supplied with fine olivine amendment, with values markedly above base concentrations in the setup with coarser olivine (Fig. 3, Fig. 5). A pronounced increase of pH at the beginning of the experiment (Fig. 4), with values near 9 for the fine grain size setup, can be observed. If values are compared against untreated mesocosms, the pH of

soil solutions increases by up to 1.0 and 0.3 pH units in mesocosms treated with fine and coarse olivine respectively (Suppl. Mat. Fig. S14-1). Over the course of the experiment, the observed pH approaches values around 8. Depending on the setup, the pH in the fine setup is about 0.5 pH units higher than in the others after one year. Si concentrations develop dissimilarly, with most pronounced increases in the coarse setup whereas the fine setup releases less than half of the Si into the soil solution in the surface level, i.e. at 1.5 cm depth (Fig. 6). The effect is less obvious in the second sampled depth at 12.5 cm and no

changes are visible below. Interestingly the Si concentrations in the top sampling for treatments with the fine material are also lower than if no olivine was supplied. With the exception of visible differences in Si concentrations, with lower values in the

setups with plants, no clear difference pattern can be identified if crop plants are present (Fig. 3 - Fig. 6 and Suppl. Mat. Figs S3-1 to S6-1). The general pattern is a large variation in concentrations suggesting that the variability between mesocosms is high and that five replicas per setup are probably not enough to derive a differentiated signal as presented for the major element concentrations. Despite the large variability, it is clear that the weathering signal from the amended olivine travels slowly downwards in the soil pore space. Within the first year, it was not moving much beyond the 12.5 cm level, as elevated Mg concentrations, which provide the clearest signal for olivine dissolution, were not clearly detectable at the third level (24.5 cm), with two exceptions in case of the fine grain setup.

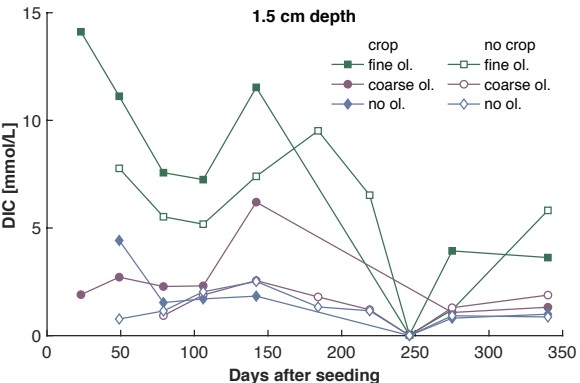

Fig. 3: Development of average DIC concentrations over one year at 1.5 cm depth, differentiated by olivine and crop treatment. For more information on the subsequent layers and error bars, please refer to Suppl. Mat. Fig. S3-1.

Fig. 4: Development of pH values (averaged proton concentrations, converted to pH) over the experiment period at 1.5 cm depth, differentiated by olivine and crop treatment. For more data and error bars, please refer to Suppl. Mat. Fig. S4-1.

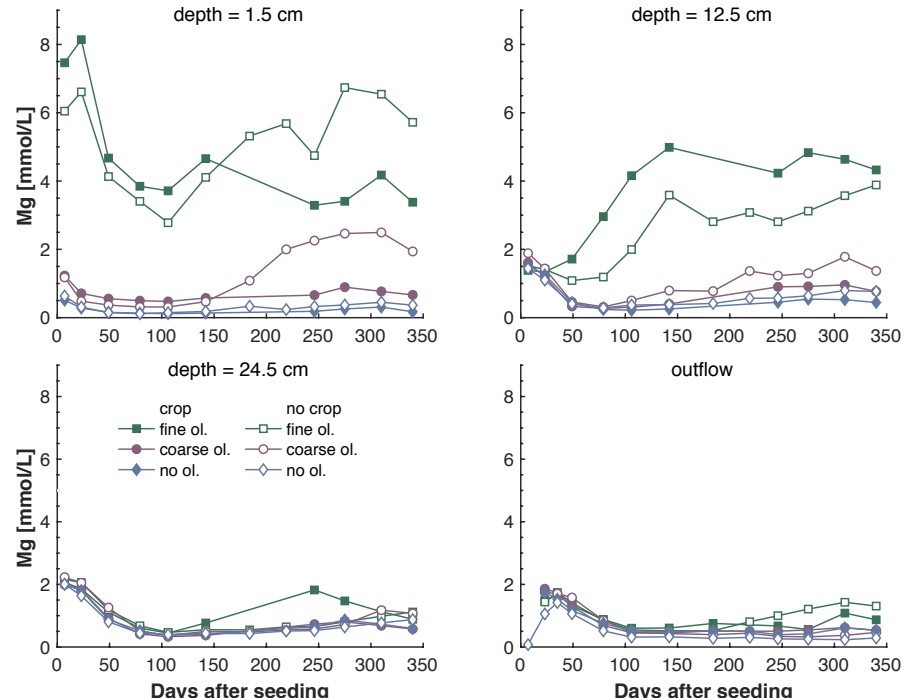

**Fig. 5: Development of Mg concentrations over the experiment period, differentiated by olivine and crop treatment. Data points are averages but error indicators were omitted to provide a better overview. For a more differentiated view and standard deviations, please refer to Suppl. Mat. Fig. S5-1.**

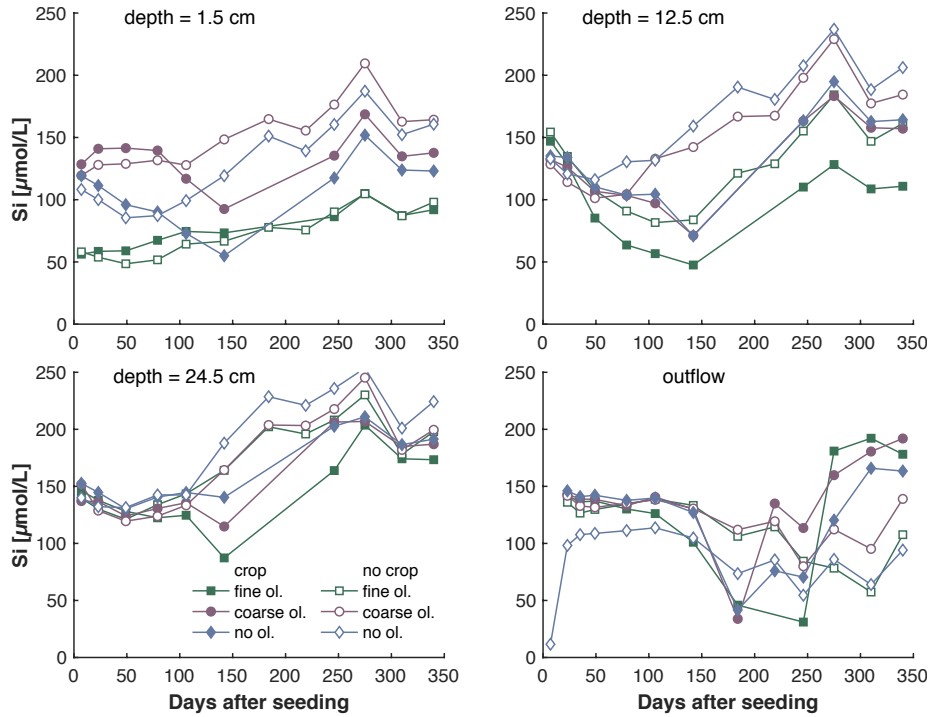

**Fig. 6: Development of Si concentrations over the experiment period in the surface layer, differentiated by olivine and crop treatment. Data points are averages but error indicators were omitted to provide a better overview. For a more differentiated view and standard deviations, please refer to Suppl. Mat. Fig. S6-1.**

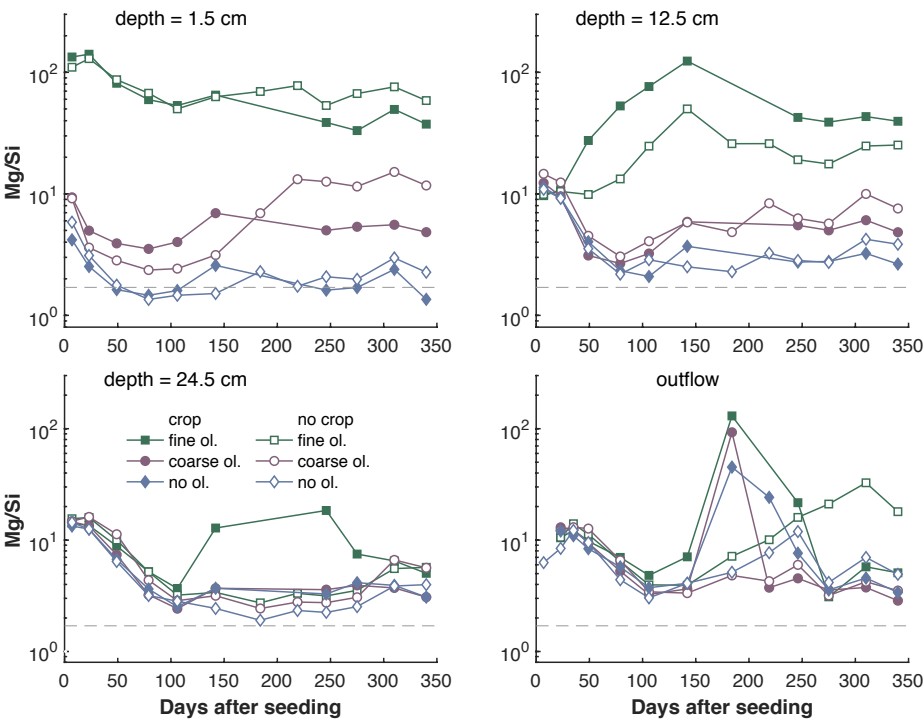

**Fig. 7: Development of Mg/Si ratios over the experiment period in the surface layer, differentiated by olivine and crop treatment. The dashed grey line indicates the stoichiometric Mg/Si ratio of 1.8 based on the rock chemistry. Data points are averages but error indicators were omitted to provide a better overview. For a more differentiated view and standard deviations, please refer to Suppl. Mat. Fig. S7-1.**

Generally, Mg/Si is clearly above 2 at the 1.5 cm and 12.5 cm level below surface in mesocosms amended with fine olivine. The ratio is roughly in the range of 1-10 in the lower sampled depths of fine treatments and in all depths of the setups without coarse and no olivine (Fig. 7). Shortly after the start of the experiment Mg/Si ratios (Fig. 7) are high (Mg/Si > 50) in the soil water at the surface of the fine grain treatment, due to a strong increase of Mg and the comparably low increase of Si. The effect is weaker for the coarse grain treatment (Mg/Si < 30, but still above 2). There is no distinct difference in Mg/Si in the three setups (fine, coarse, no olivine) in the deepest soil sampling point and the outlet (with exception of two outlier points in the deepest sampled layer for the fine olivine with crop setup).

## 3.3 CO₂ sequestration rates

Ideally, the $CO_2$ consumption by weathering can be calculated based on DIC or alkalinity. As too few samples were available for DIC analysis, the additional $CO_2$ consumption by olivine amendment was calculated based on the release of $Mg^{2+}$, considering the average geochemical composition of the material and the background values from applied soils and irrigation water. Based on the stoichiometric composition, the ideal dissolution of 1 mol Mg-olivine yields 2 mol Mg and consumes 4 mol $CO_2$:

$$Mg_2SiO_4 + 4H_2O + 4CO_2 \rightarrow 2Mg^{2+} + 4HCO_3^- + H_4SiO_4$$

The ability to sequester atmospheric $CO_2$ is material specific and depends here on the $Mg^{2+}$ that can be released during hydrolysis from the Mg-rich olivine. It is defined as the carbon dioxide removal $RCO_2$ in tons of $CO_2$ per ton of Mg-olivine (estimated to be 1.1 for ultramafic (i.e. Mg rich) rocks (Moosdorf et al., 2014)). This assumption is considering that impurities (like Fe abundance) in contrast to the ideal Mg-olivine and equilibration effects reduce the theoretical maximum $RCO_2$ of 1.25 for forsterite. Based on average of Mg concentrations in the outlet water over the first year (340 days), the experiment leads to a total annual $CO_2$ sequestration of 2.3 - 4.9 t $CO_2$ $km^{-2}$ $a^{-1}$, depending on the applied grain size (Tab. 3).

To evaluate the potential order of magnitude of Mg uptake or dilution, possibly introduced by the experimental setup, Mg concentrations in the outlet water might be compared to those in the surface layer pore water. The ratio of surface layer Mg concentration to outlet Mg concentration, is 12.2 in the coarse setup and 13.8 in the fine setup (Tab. 3).

Tab. 3: Mg and water flux averages (± SD) throughout the period of the experiment, excluding background contributions from soil and irrigation water, for crop and no crop treatment. The potential of Mg removal and its effect on $CO_2$ consumption is provided, assuming that all water would percolate the pore space of the upper soil and equilibrate towards the measured Mg-concentrations. Details on the calculation of $CO_2$ consumption are found in section 2.4.

| | Mg [$\mu$mol $L^{-1}$] | | Mg reduction ratio[a] | Water flux at outlet [L $d^{-1}$] | $CO_2$ consumption [t $CO_2$ $km^{-2}$ $a^{-1}$] | |
|---|---|---|---|---|---|---|
| | surface layer | outlet material | | | "observed"[b] | potential[c] |
| fine | 4713.1 ± 1128.2 | 357.8 ± 238.1 | 13.8 ± 7.4 | 0.8 ± 0.6 | 4.9 | 67.6 |
| coarse | 918.5 ± 534.6 | 79.0 ± 30.0 | 12.2 ± 6.6 | 0.8 ± 0.6 | 2.3 | 28.0 |

[a]Calculated directly from original Mg concentrations per sample time step (not averages, hence the slight difference to the ratio of average Mg concentrations from the first two columns). Data were taken only from day 79 onwards because fluctuations were too inconsistent in the first weeks of the experiment. [b]Calculated net flux from mesocosm based on observed Mg concentrations in the outflow. [c]$CO_2$ consumption "observed" at outlet multiplied with reduction ratio to account for Mg removal or dilution.

## 3.4 Trace metals

### 3.4.1 Soil

Analyses of the soil elemental composition shows that some trace element concentrations are elevated, where olivine was applied (Fig. 8). Markedly, this is the case for Co, Cr, Ni, Mn, Al and Fe. There are no statistically significant differences between the applied grain sizes and crop types.

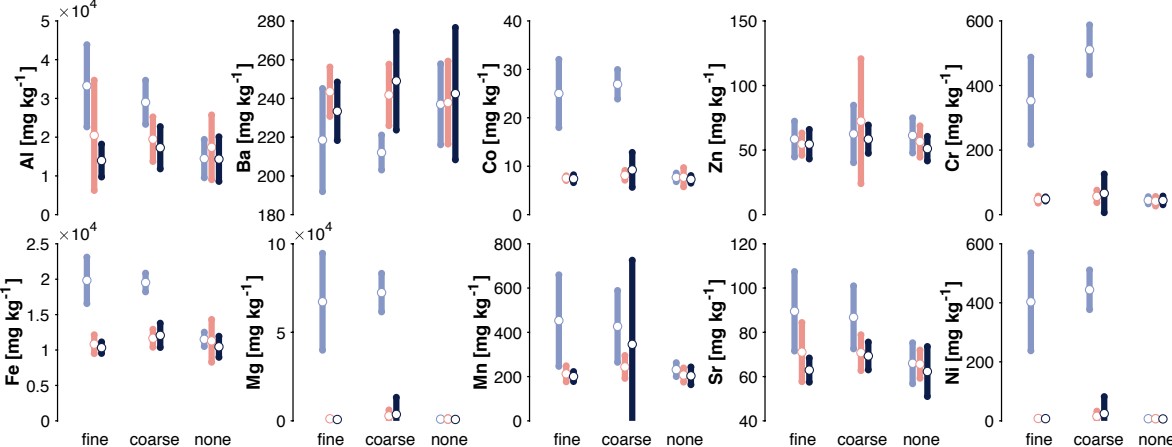

**Fig. 8: Averaged trace element concentrations in the solid soil material, differentiated by olivine treatment (grouping) and depth (blue: 2-4 cm, red: 10-12 cm, black: 18-20 cm below surface). Error bars indicate 1 standard deviation. Data separated by plant type in Supplement Mat. S8 (topsoil only).**

### 3.4.2 Soil solution

Ni and Cr concentrations in the soil solution are elevated in the surface layer where fine olivine grain sizes were applied (Fig. 9) during the first 100 days. The coarse grain setup does not show any visible Cr concentration difference compared to the control, Ni concentrations in barrels amended with olivine are higher than in the mesocosms without olivine on average (Fig. 9). The existence of plants does not cause a distinct pattern change in any setup compared to no plant treatments. Base values of Cr without olivine treatment are already above 50 nmol $L^{-1}$. The dissolution of dunite predominantly leads to elevated levels of Ni and Cr concentrations in the soil solution over the control (Suppl. Mat. Fig. S11-1.).

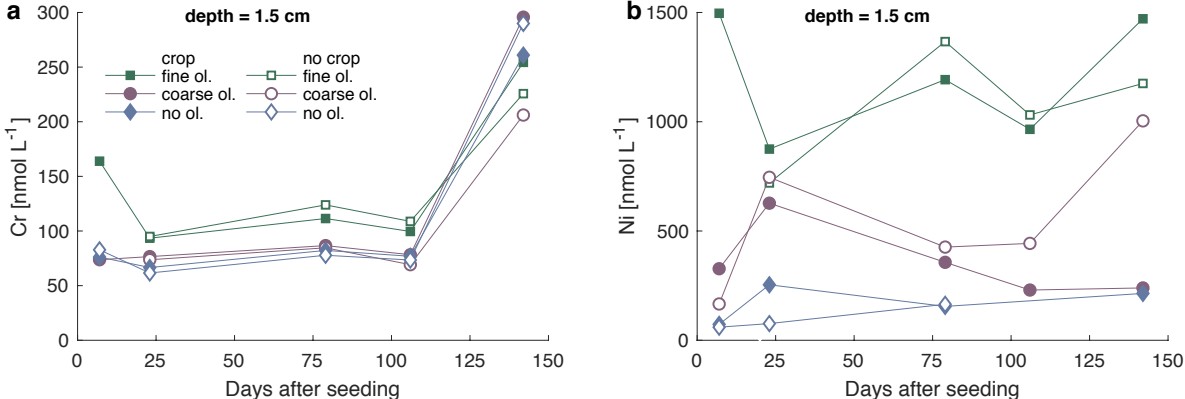

**Fig. 9: Development of Cr (a) and Ni (b) concentrations in soil pore water over the first 5 months in the layer 1.5 cm below surface, differentiated by olivine and crop treatment. For more data and error bars, please refer to Suppl. Mat. Fig. S9-1 and S10-1.**

For the other trace elements, no distinct pattern between treatments with and without olivine were identified, apart from a general variability found in the solutions.

## 4 Discussion

### 4.1 Tracing the weathering effect

Based on the released Mg and BET surface area measurements of the ground rock material, weathering rates can be estimated (cf. section 2.4). Derived rates of $10^{-13.12}$ and $10^{-13.75}$ mol Ol m$^{-2}$ s$^{-1}$ for coarse and fine material respectively, based on the outlet water, are about a magnitude lower than values published for a olivine amended soil column experiment (Renforth et al., 2015) and about three orders of magnitude lower than theoretical optimum dissolution rates given in Strefler et al. (2018). These differences are not unexpected, considering that Mg is not acting conservatively, and our experimental setup simulates natural processes like extended periods of drying out, with potential secondary mineral formation, and subsequent slowed down or ceased chemical weathering processes, as well as cation exchange, and dilution by preferential flow.

The large difference of available surface area for reaction between fine and coarse material (Tab. 2) has implications for the release rate of elements as it is proportional to the available surface area. Averages of Mg, with the clearest dissolution signal, show a strong difference between Mg/Si release ratios of the fine and coarse material setups, suggesting that the dissolution of the dunite and predominantly olivine is clearly not stoichiometric. This was also observed in results from laboratory experiments (Pokrovsky and Schott, 2000). We assume that the potential formation of cation depleted silica layers around minerals might affect the dissolution rates. Further effects are related to the distribution of water in the pore space, which is steered by grain size distribution effects, via differences of the water contact time with grain surfaces. Also, climatic conditions lead to drying-up during the warm period in the greenhouse caused by evapotranspiration under presence of plants, resulting in low soil water content and varying elemental concentrations in the remaining water. If the Mg/Si ratio is two, the ideal stoichiometric molar ratio of element release from forsteritic olivine is reached. In the case of the applied material, the Mg/Si ratio is about 1.8 (inferring from RFA results). The experimental data show that Si and Mg concentrations in the upper layer are often decreased if plants were present (Fig. 5, Fig. 6). But even with this effect, Mg/Si ratios are still far from the equilibrium release ratio of 1.8. This effect is widely recognised as incongruent dissolution (Casey et al., 1993; Ruiz-Agudo et al., 2012). Considering that there are large amounts of Mg released, Si determines the ratio assuming that removal by plants is minor. This suggests that there is an active retention of Si, potentially leading to a cation depleted amorphous silica layer growing around olivine minerals (Daval et al., 2013b; Daval et al., 2011; Hellmann et al., 2012). This effect has been described in detail for forsterite by Maher et al. (2016). High Mg/Si ratios indicate that in the beginning of the experiment the dissolution rate is controlled by the exchange of protons for Mg, while declining ratios over the course of the experiment indicate an approach to steady-state conditions (Maher et al., 2016). Since this effect may eventually determine the $CO_2$ sequestration rate, an estimation of the thickness of those layers would lead to a better understanding of weathering kinetics for silicate application schemes in general, as this process may affect other silicate minerals too. From the mesocosm experiments, it is only feasible

to calculate a rough first order estimate given the rather basic setup of the mesocosms. Calculated amorphous layer growth rates (details section 2.5) for the mesocosms without plants range from 0.02 nm a$^{-1}$ (fine setup) to 0.08 nm a$^{-1}$ (coarse setup). These values are above or near observations of surface layers around "aged" minerals, e.g. from Hellmann et al. (2012): 14.7 ka old feldspar with a surface layer of 50 nm ($\approx 0.0034$ nm a$^{-1}$) and a layer of 150-200 nm on a younger (assuming 10 ka)

serpentine ($\approx 0.02$ nm a$^{-1}$). However, if assuming that the fresh surfaces of the forsterite are weathering faster in the beginning and a decrease in reaction rate can be caused by the formation of the amorphous layer due to processes related to the diffusion of released elements through the layer (Nugent et al., 1998; Daval et al., 2013b), this might explain why the calculated growth rates are comparable to aged material, as with time, weathering rates decrease.

As the formation of a secondary layer through Si reprecipitation should preferentially include Si previously released by the

amended material, this process alone cannot explain why Si concentrations in the fine setup are about twofold below values of the control setup without olivine and without plants (Fig. 6). We speculate that the increased release of Si from the finest grains leads to short term oversaturation of Si (about 1.9 mmol L$^{-1}$ at 25°C; Stumm and Morgan (1996)), extending beyond the typical area of the secondary silica layer formation around the grains, which may facilitate the formation of secondary minerals such as smectites (Prudêncio et al., 2002) or a mixture of different hydrous silicates of iron and magnesium, known as iddingsite

(Smith, 1987).

## 4.2 CO₂ sequestration by olivine amendment

The pH increase, as an indicator of rock dissolution of silicates is most pronounced in the upper layer and during the first six months of the experiment. This indicates an enhanced reaction with the added rock powder, which contains a large fraction of very fine material, providing an increased reactive surface area. The effect indicates the generation of alkalinity by chemical

weathering consuming $CO_2$ and can partly be seen in the DIC concentrations for the fine grain experiments (Fig. 3). As DIC was handled with the lowest priority during the sampling campaign (regarding low sampling volumes), only a few measurements are available which makes it hard to truly differentiate the treatments based on DIC.

Due to limits in the acquired data, it is only possible to give a rough estimate of the $CO_2$ drawdown effect by weathering. The elevated elemental concentrations from the surface sample point do not progress evenly downwards, which means that several

processes influence Mg concentrations during downward percolation. A Mg concentration difference between surface soil pore water and outflow by about an order of magnitude could primarily stem from element removal by plants, mineral precipitation, de-mobilization by cation absorption through clay minerals or dilution through preferential flow along soil macropores and the rims of the mesocosm barrel. Significant and systematic Mg removal by plants might be ruled out, as differences between Mg concentrations of the surface pore waters in planted and unplanted mesocosms are mostly not

significant (SI Fig. S5). As Mg in the surface layer and the outflow increase in nearly all treatment types when evapotranspiration values rise due to increasing temperatures from around day 100 on, a process also observed in another mesocosm experiment (Thaysen et al., 2014), the loss of Mg by secondary mineralization can be assumed but not quantified. However, the precipitation of Mg-carbonates at ambient conditions is kinetically hindered (Case et al., 2011; Giammar et al.,

2005) and therefore, the Mg removal effect is potentially small for carbonates in the mesocosms environment, but further investigation for amorphous phases or precursors for secondary mineral formation would be necessary.

The soil is characterized by an average CEC of 8.6 meq (100 g dry soil) $^{-1}$. While the value is comparably low (Batjes, 1997), it can be expected that Mg is adsorbed onto surfaces of clay minerals and organic matter at the given elevated pH values.

It is furthermore possible that some parts of the irrigation water are bypassing the bulk material as preferential flow, possibly along the barrel's rims or potentially through the soil, facilitated by plant roots or evoked by macropores, which lead to the preferential transportation of water downwards, effectively decreasing rock-water interaction times (Nielsen et al., 1986) and therefore dissolution rates in the affected material (Maher, 2011). The process of preferential flow is well established for natural soils (Beven and Germann, 2013), and we assume it asserts some influence on the outflow elemental concentrations in

this mesocosm experiment. Future Enhanced Weathering experiments using mesocosms or field experiments may therefore include tracking the effect of preferential flow-paths, as it seems from this experiment that hydrology maybe a factor introducing large uncertainty, considering also the seasonality and periods of drying affecting the concentrations and the flux from the system (c.f. Fig. 2 with SI Fig S12-1).

The amount of Mg released can theoretically be estimated by multiplying its concentrations with the calculated average outflow

water volume at the mesocosm bottom (Suppl. Mat. Fig. S12-1). Yet, due to the observed difference between surface layer and outflow Mg concentrations, it must be assumed that only a small proportion of the initial reaction with the applied grains eventually leads to $CO_2$ sequestration. While this Mg removal may be specific to the experimental setup, there will be comparable effects in natural environments. To determine the potential order of magnitude of this effect, Mg concentrations in surface layer pore water and outlet water were compared and the ratio is in both setups near 13 (Tab. 3), indicating that the

$CO_2$ uptake potential, measured in dissolved Mg, is decreased by more than 90% compared to the surface layer. The estimated total annual $CO_2$ sequestration of maximum 4.9 t $CO_2$ km$^{-2}$ a$^{-1}$ is two orders of magnitude lower than what was observed in a soil column of 10 cm diameter and without plants (Renforth et al., 2015). Applied at the global scale, i.e. on all potentially available arable land (min/max taken from Moosdorf et al., 2014), this yields a comparably low $CO_2$ sequestration potential of maximum 0.07 Gt $CO_2$ a$^{-1}$, or less than 0.2% of the global fossil $CO_2$ emissions of 2017 (based on Le Quéré et al.,

2018).When the calculations are based on a Mg flux neglecting the obvious removal or upconcentration of Mg due to evapotranspiration, values are about one order of magnitude higher (Tab. 3), which makes them comparable to lower end sequestration rates reported from a smaller pot experiment in ten Berge et al. (2012), which mainly was watered at the bottom to avoid downward percolation, thus leading to a different environment for elemental cycling.

Also, the globally upscaled estimates do not take geographic variability into account. Since weathering rates are elevated in

humid (sub-)tropical regions, the global potential based on temperate conditions, and given a comparably low amount of rainfall, is clearly underestimated. Data on $pCO_2$ in the mesocosm soils (Suppl. Mat. Fig. S13-1) corroborates that weathering effects must be less pronounced compared to humid, tropical areas as values in the experiment were 850-1300 µatm in the surface layer (depth = 5 cm) whereas they are up to thirty times higher in areas with high evapotranspiration (Brook et al., 1983). Soil water content also seems to be an important control on the soil-rock $PCO_2$ (Romero-Mujalli et al., 2018), and

therefore seasonality controlling soil water content is likely to be a relevant factor influencing the dissolution kinetics, via the control on soil $pCO_2$, being an important agent in the dissolution process of minerals. Overall, the assessment of global potentials by mesocosm experiments requires setups simulating humid (sub)tropical conditions, which would promote larger fluxes through the soil column.

The large stretch of results shows a) that cation removal by different chemical and physical processes in soils is an important parameter to include in flux estimates at larger scales, which was not considered in detail so far, b) that it is fundamental for such an experiment setup to monitor the water- and elemental fluxes through the entire mesocosm, and c) that the role of seasonal dynamics and amount of the weathering agents like $CO_2$ in the soil should be considered.

## 4.3 Trace elements and processes

The release of trace metals which can be potentially harmful to the environment was mentioned as one of the side effects of terrestrial Enhanced Weathering (Hartmann et al., 2013). The effect is especially pronounced when rocks like the dunite in this experiment are applied, since they contain larger amounts of Cr and Ni (>2000 ppm each, Tab. 1). Nickel (Ni) in olivine is expected to be released as it substitutes for Mg. Chromium (Cr) on the other side, here present in chromite and chromochlorite from the dunite source rock, is not expected to be released strongly at the observed pH levels in the system.

Soil analysis confirms an increase of these trace metals, due to the added material. The data also shows that the lower untreated layers are little affected by the dunite treatment (Fig. 8). Focussing on Ni and Cr, which are the predominant trace metals in the applied material, soil solution concentrations fluctuate strongly (Fig. 9) due to warmer periods, and subsequent drying-out, causing enrichment of dissolved elements in the solution.

Ni is partly mobile at the given pH values. Observed Ni concentrations exceed drinking water quality thresholds in the surface
layer, e.g. formulated by the WHO (2011) with 0.02 mg $L^{-1}$ ($\triangleq$ 339.2 nmol $L^{-1}$), yet is within the recommended limits for agricultural irrigation water (0.2 mg $L^{-1}$, Ayers and Westcot, 1985). This demonstrates that a close water monitoring is necessary to understand implications of a widespread deployment of Enhanced Weathering with source materials containing such elevated concentrations of mobile trace metals. Avoiding these particular rock types materials might therefore be the best alternative.

When comparing the theoretical Mg/Ni ratio in the olivine with measured data in the soil solution of the surface layer (Tab. 4), it can be seen that less Ni is in solution than theoretically possible (a factor of 10 – 20 difference, depending on the grain size). Under the given physicochemical conditions it is possible that Fe- and Al-Hydroxides lead to the partial sorption of Ni (Young, 2013; Rieuwerts, 2007).

**Tab. 4: Preferential release of Mg over Ni into pore water solution during the dissolution of dunite. Molar ratios of Mg/Ni in the dunite are based on XRF analysis. Surface layer solution values for fine/coarse setups are averaged over the experimental period.**

| | rock | surface layer solution | |
| | | fine setup | coarse setup |
| | [mol kg$^{-1}$] | [µmol kg$^{-1}$] | [µmol kg$^{-1}$] |
| --- | --- | --- | --- |
| Mg | 11.2 | 4175 | 638 |
| Ni | 0.05 | 1.03 | 0.26 |
| Mg/Ni molar ratio | 224 | 4053 | 2454 |

At the same time, Cr is apparently less mobile (Suppl. Mat. Fig. S11-1) in the mesocosm experiment, considering the sample approach, which is matching the general behavior of Cr mobility at pH values measured in the soils (at pH values of 7-9; Kabata-Pendias (1993)). However, elevated Cr values have been shown for a column experiment, which is to some extent comparable to the mesocosm experiment here. In contrast to our experiment, Cr values increased stronger by up to 9 ng g$^{-1}$ ($\approx$ 173 nmol kg$^{-1}$), including background Cr (Renforth et al., 2015). Interestingly, Cr seems to be actively removed from the solution at a later stage of the experiment, shown by Cr concentrations in the untreated mesocosms being higher than in the treated mesocosms (Fig. 9a), probably due to the higher pH compared to the control, supporting the idea of pH management to immobilize Cr.

If trace elements within the applied dunite remain immobile, they accumulate in the soils and can potentially be released when the pH is dropping or redox condition change (McClain and Maher, 2016 and references therein). Grain size effects are visible, shown by elevated concentrations of Ni in the mesocosms amended with fine olivine. Other trace elements are represented only in smaller amounts in the composition of the source rock, thus, not releasing relevant amounts into the pore water.

Overall these findings from the mesocosm experiment underline the proposition to focus on alternative sources like basalt (Hartmann et al., 2013; Taylor et al., 2015; Strefler et al., 2018; Amann and Hartmann, 2019) to avoid strong environmental impacts from trace element release.

## 5 Conclusion

Given the scarcity of data considering the field application of rock material for Enhanced Weathering, or compilation of research with other purpose, there are some lessons to be learned from this experiment. The elevated Mg concentrations indicate the potential of an inorganic $CO_2$ sequestration effect, and the order of magnitude is possibly large enough for the method to be considered to be one piece in the puzzle of negative emission technology portfolios. However, the calculations are bound to high uncertainties mainly from water flow and elemental concentration estimates throughout the mesocosm. It is crucial to understand processes that can affect the $CO_2$ sequestration potential or impact its assessment, i.e. evapotranspiration, sorption of cations, secondary mineralization, or low overall residence time due to preferential flow of irrigation or rain water. The concentration difference between the upper soil and the outlet water by about a magnitude indicates the relevance to include those processes in future experiments to parameterize standardized flux calculations, specifically for environmental

conditions with promising $CO_2$ sequestration potential by Enhanced Weathering in areas more humid than represented by the setup of this experiment. This is especially relevant if the Enhanced Weathering $CO_2$ sequestration was to be coupled to a carbon price to regain application costs (Hartmann & Kempe, 2008). Further beneficial services by Enhanced Weathering like the potential increase in plant biomass via uptake of growth-limiting elements provided by rocks, or the increase of secondary

minerals positively affecting nutrient retention, was not investigated here but is an essential part of future studies to assess the full potential of Enhanced Weathering as a method for carbon dioxide removal (Amann and Hartmann, 2019), and potentially for pH management avoiding release of further greenhouse gases like $N_2O$ (Kantola et al., 2017).

One of the main concerns of the rock powder application is the release of potentially harmful trace elements. It could be shown for the first time in a dedicated Enhanced Weathering experiment in an open system with plants, that levels of Ni in solution

are significantly elevated whereas it was possible to confirm that Cr levels in solution are low under the given soil conditions. The experiment also showed that the behavior of silica in the soil is not well understood if silicate powder of different grain sizes is applied. This is evidenced by the high Mg/Si ratios and the potential sink of silica in comparison to the non-silicate treated mesocosms. Results appear to be consistent with published observations that the formation of a cation depleted and Si enriched grain surface layer, is responsible for the missing silica. The available data do not allow further conclusions here.

Nonetheless, the effect of a growing depleted silica layer on the dissolution kinetics and of further secondary mineral phases, should be investigated, specifically if a long-term application of Enhanced Weathering is envisaged. Using more complex rock products, like basalt with higher aluminum content, may produce larger amounts of new phyllosilicates and other products around the added fresh mineral grains, changing their kinetic behavior in the long-term.

Overall, this shows that mesoscale and field experiments are of utmost importance to identify the essential processes, to

decrease uncertainties in process understanding, element releases, and to address the effects of elevated element fluxes. Only if budgets of Enhanced Weathering can be estimated reliably, the resulting $CO_2$ consumption could be bound to a carbon prize within a NET deployment strategy.

**Competing interests**

The authors declare that they have no conflict of interest.

**Author contribution statements**

This article was conceived by the joint work of E.S., J.S., J.H., and T.A., which all participated in discussions, planning and writing, with the lead of T.A.. The mesocosm study was conceived and designed by E.S., J.S., P.M., and I.J.. Sampling was primarily conducted by E.S. and J.S.. Mg and Si analyses were done by E.S. and J.S., trace elements were analyzed by E.K.F., DIC was measured by T.A.. W.O.G. contributed to the discussion of trace elements. J.H. contributed to the discussion of

weathering effects.

## Acknowledgments

This research was executed with the financial support of the Research Foundation Flanders (FWO), project no. G043313N 'Silicate fertilization, crop production and carbon storage: a new and integrated concept for sustainable management of agricultural ecosystems'. J.S. is a postdoctoral fellow of FWO (project no. 12H8616N). Additional support was provided by the German Research Foundation's priority program DFG SPP 1689 on "Climate Engineering–Risks, Challenges and Opportunities?" and specifically the CEMICS2 project to T.A., J.H. and W.O.G.. Further support to T.A. and J.H. came from the Deutsche Forschungsgemeinschaft (DFG, German Research Foundation) under Germany´s Excellence Strategy – EXC 2037 'Climate, Climatic Change, and Society' – Project Number: 390683824, contribution to the Center for Earth System Research and Sustainability (CEN) of Universität Hamburg, and through the previous EXC177 'CLISAP2', Universität Hamburg.

We acknowledge Peggy Bartsch, Tom Jäppinen, Marvin Keitzel, and Andreas Weiss for valuable contributions from the wet lab, and Sebastian Lindhorst for providing granulometric analyses (all from Institute for Geology, Universität Hamburg). We thank Stephan Jung and Joachim Ludwig (from Institute for Mineralogy and Petrography, Universität Hamburg) for contributing the XRF and XRD analyses. All employees of the Antwerp City greenhouse are thanked for their practical support.

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
