# Peer review of "Constraints on Enhanced Weathering and related carbon sequestration – a cropland mesocosm approach"

_Biogeosciences, 2018_

## Short Comment (SC1) · 6 Nov 2018

I have the following considerations: 1. Include the soil material characteristics used in the mesocosm in the main MS. important would be soil C content, soil pH value (in CaCl2) and texture. The origin (horizon of which WRB soil type) would also be of interest. It can be assumed that an acid soil respond differently than an alkaline soil. 2. Please specify the amount of olivine for suitablity under field conditions. The 22 kg m-2 corresponds to 220 t ha which seems very high, or? 3. Can you also add a direct evidence of C accumulation during the period, formation of C org and C inorg? 4. How was the productivity of the plants affected by the olivine application?

---

## Referee Comment (RC1) · Jessen (Referee) · 4 Dec 2018

Amann et al. analyse results from a well performed mesocosm experiment. The experiment is apparently mainly aimed at deducing the rate of terrestrial carbon sequestration from soil amendments by dunite (olivine). The manuscript clearly addresses scientific questions of relevance for Biogesciences journal.

Below I list general comments. However, please see also the attached PDF-file, which contains comments of both major and minor importance. My comments in this file and below may be perceived as very direct; therefore please allow me to express my deepest respect for the authors, their work, and their important choice of subject. I

truly believe the manuscript can become an important and significant contribution to the literature!

Yours sincerely
Soren Jessen

General comments

1. A set of $CO_2$-sequestration rates makes use of a correction for preferential flow. I find this problematic. The underlying reasoning for factoring up the inferred $CO_2$-sequestration is that lots of weathering take place in the statically held back solution, which is allegedly continuously bypassed by macropore flow.

However, even in the case of a dominantly preferential flow system, the net weathering rate would still be a function of the effluent water flux multiplied by its weathering product concentration and a stoichiometrical coefficient. The deduction of a preferential flow factor therefore is, as far as I am convinced, purposeless for estimating the $CO_2$-sequestration.

Furthermore, elsewhere in the manuscript the authors state that "extended periods of drying out" resulted in "slowed down or ceased chemical weathering processes", which is quite the opposite to the assumption made above of quantitatively important reactions in zones of static water. So there is a contradiction here which also needs to be solved.

2. The manuscript concludes that $CO_2$-sequestration "was shown" (without the above correction). I disagree: DIC needs to percolate to the groundwater table and beyond, and/or form (stable) carbonate minerals and/or org. C stocks. The latter two are not measured, and effluent DIC concentrations are (according to the manuscript) measured too infrequently to be applicable. The use of Mg as a proxy for DIC need to be carefully documented by data (not just by theoretical reaction stoichiometry) before it

can be used as direct evidence. I think the authors should include a discussion (in the Introduction) of the requirements for enhanced weathering to actually be achieved.

In my opinion, the 'fixes' for the manuscript to 1) and 2) above might include to acknowledge that the elements under consideration (e.g., Mg) are non-conservative, and that the retarding processes are not investigated mechanistically in this study (at least, the data were not shown). Therefore the preferential flow-calculation (which uses Mg) must be skipped, and the tight conclusion regarding the sequestration most be softened. While the sequestration cannot be said to be "shown", in my opinion, I do think the authors could safely say that their results 'indicate a potential' $CO_2$-sequestration of X t C/ha/yr.

3. A thorough analysis of the water balance for the mesocosms needs to be presented. The water balance need to include an evaluation of the transport time for water through the mesocosms.

4. The overall purpose of the study appears to be slightly blurred. The focus on estimating $CO_2$-sequestration rates infers that this was the main aim. (However, one must then ask why DIC in the effluent was not more carefully measured, ie., what were the mesocosms designed for?)

Another purpose appears to be to demonstrate the use of dunite as a "model mineral" for enhanced weathering experiments. (But then, why the strong focus on the estimation of $CO_2$-sequestration rates and trace elements, which implies a focus on field applicability?)

The authors should state the objective(s) more clearly.

5. Presentation and structure: The manuscript needs to undergo a major revision in terms of the structure, conciseness of the text, and its figures. For example, the manuscript contains many repetitions and many imprecise statements. Also, many results are presented in the Discussion and some discussion take place in the Results

section. Some results were not presented (or did I miss them during my reading?), but were still referred to/used in the Discussion. The artwork need to be polished; generally, the figures in the supplementary information seem to be better worked through than the figures in the manuscript, although the depths (e.g., cm below soil surface) needs to be added in the supplementary material, rather than using a 'depth number'. Five replicates for each treatment combination were conducted, but this need to be visualized by statistics in the figures.

Please also note the supplement to this comment:
https://www.biogeosciences-discuss.net/bg-2018-398/bg-2018-398-RC1-supplement.pdf

**Supplement:**

[revised manuscript text omitted]

---

## Referee Comment (RC2) · Manning (Referee) · 17 Dec 2018

I like and welcome this paper. I think we need data from experiments of this type, which represent a major investment of researcher time, to thoroughly understand enhanced weathering. The data that are presented are really useful.

I have a few comments that in some cases could be included in a final revision, or they can enter a discussion.

1) 22kg/m2 (page 4) is a very high application rate! The plants will be growing in olivine. Is this the correct application rate or has a gremlin affected the units?

[Figure]

2) I note you state the source of the olivine. Are there any references to other published descriptions of the material in the Åheim deposit? It would be good if these could be provided, as you only give basic information concerning the mineralogy and geochemistry.

3) With that in mind, a good reference to the deposit might address these queries concerning Table 1: a) why is the LOI so high?; b) could you recalculate the mineral composition that the chemical analysis represents?; c) is any asbestos associated with this material?; d) total iron is given as Fe2O3 (this should be stated), yet olivine contains divalent iron. What is the iron mineral in this material?

4) Some typos: p4 line 15 - X-ray not x-ray; p5 line 10: magnesium not Magnesium; p6 line 21: through not trough. Check once more for other typos elsewhere!

5) Back to the science: do you have Si and Mg data for the plant biomass? I think this is important to give a mass balance of removal of these elements from the soil and its constituent minerals.

6) Did you find any evidence of precipitation of Mg carbonate minerals, as reported for 'similar' rocks by Dipple's group? Did you look for these minerals?

7) Both Cr and Ni are essential nutrients for a range of biological processes. I'd prefer to avoid the use of the emotive word 'contaminant' as we'd all be dead without sufficient of these elements in our diet.

8) Again, do you have any evidence for differential uptake of these elements into the crops? If there is no significant difference between treatment and control, then you have no evidence of a problem.

9) What was the mineralogical composition of the soil that was used? This should be stated, to ensure that any confounding factors (such as preferential weathering of a soil mineral already there) can be assessed. I appreciate the design of the study would avoid such factors, but it would be very useful to know. For example, does the soil

contain carbonate minerals?

---

## Author Comment (AC1) · 29 Mar 2019

Dear Oliver, Thank you for your comments. In the following we document our replies and changes that we made according to your suggestions.

**Reviewers comment**

Our reply

**1. Include the soil material characteristics used in the mesocosm in the main MS. important would be soil C content, soil pH value (in CaCl2) and texture. The origin (horizon of which WRB soil type) would also be of interest. It can be**

[Figure]

**assumed that an acid soil respond differently than an alkaline soil.**

We included all available data on the soil in the supplement section S1.

**2. Please specify the amount of olivine for suitablity under field conditions. The 22 kg m-2 corresponds to 220 t ha which seems very high, or?**

Yes, the amount is high, yet leaves no visual impact on the soils. It is hard to estimate a suitable amount, since it strongly depends on the grain size distribution of soil type and rock powder. We added a small remark on the value of the application rate to avoid confusion.

**3. Can you also add a direct evidence of C accumulation during the period, formation of C org and C inorg?**

In Fig. 3 of the MS, we show the development of DIC in the soil water. Beyond this, we don't have information on carbonate precipitation or Corg formation.

**4. How was the productivity of the plants affected by the olivine application?**

Like stated in the MS, the effect of the olivine application on plant growth and productivity is the topic of the original study for which the experimental setup was designed. Results are processed in another MS that is currently being finalized. We can therefore not include them in the present MS. Moreover, the present MS focusses on the weathering rates and potential for carbon sequestration and adding information about the plants would distract from the main focus. It will also elaborate the paper considerably (more material and methods, more figures, more discussion points, etc.) which is not desirable and is the main reason for us to divide the results in 2 publications.

---

## Author Comment (AC2) · 29 Mar 2019

Dear Søren, we are grateful for your extremely thorough review of our manuscript. In the following we document the changes that we made according to your suggestions, based on the text that you wrote. Annotations that you made in the original manuscript were considered and changed directly without additional documentation here, unless we found it to be necessary to be explicitly addressed.

**Reviewers comment**

Our reply

[Figure]

**1. A set of CO2-sequestration rates makes use of a correction for preferential flow. I find this problematic. The underlying reasoning for factoring up the inferred CO2- sequestration is that lots of weathering take place in the statically held back solution, which is allegedly continuously bypassed by macropore flow. However, even in the case of a dominantly preferential flow system, the net weathering rate would still be a function of the effluent water flux multiplied by its weathering product concentration and a stoichiometrical coefficient. The deduction of a preferential flow factor therefore is, as far as I am convinced, purposeless for estimating the CO2 -sequestration.**

This is correct for the net flux from the mesocosm in this experiment. However, we point out, that this is a relevant process to be considered in nature, possibly biasing results from field experiments if trials are compared to controls. We tweaked the wording in the direction that we don't want to publish "corrected" CO2 sequestration rates, but that we can learn from the experiment, that there is a process potentially influencing the results, which is specifically important if those are used to estimate method potentials or even CO2 sequestration values for CO2 capture rewards.

**Furthermore, elsewhere in the manuscript the authors state that "extended periods of drying out" resulted in "slowed down or ceased chemical weathering processes", which is quite the opposite to the assumption made above of quantitatively important reactions in zones of static water. So, there is a contradiction here which also needs to be solved.**

We think by modifying the text according to your remarks above, this point is implicitly addressed.

**2a. The manuscript concludes that CO2-sequestration "was shown" (without the above correction). I disagree: DIC needs to percolate to the groundwater table and beyond, and/or form (stable) carbonate minerals and/or org. C stocks. The latter two are not measured, and effluent DIC concentrations are (according to**

**the manuscript) measured too infrequently to be applicable. The use of Mg as a proxy for DIC need to be carefully documented by data (not just by theoretical reaction stoichiometry) before it can be used as direct evidence.**

Mg was compared to controls, so only Mg from applied material is released and used to estimate $CO_2$ consumption. To improve the chemical background understanding, we added a weathering reaction formula for forsterite in section 3.3 and some points in the introduction, addressing your next remark at the same time.

**I think the authors should include a discussion (in the Introduction) of the requirements for enhanced weathering to actually be achieved.**

The dissolution of silicates stores $CO_2$ in form of alkalinity in oceans on longer timescales. This process previously mentioned somewhat indirectly and is now elaborated a bit in the introduction.

**2b. In my opinion, the 'fixes' for the manuscript to 1) and 2) above might include to acknowledge that the elements under consideration (e.g., Mg) are non-conservative, and that the retarding processes are not investigated mechanistically in this study (at least, the data were not shown). Therefore, the preferential flow-calculation (which uses Mg) must be skipped, and the tight conclusion regarding the sequestration most be softened.**

Yes, there is a potential influence of plants and other processes (removal by cation exchange and precipitation). However, we try to give a first order estimate for the process of preferential flow to point out its potential to change the impact of $CO_2$ sequestration or weathering estimates. Rebuttal Fig. 1 (reply to your remark 5c) shows that if we add the error bars (standard deviations) the differences between crop and no crop treatment are not significant. Nonetheless, we changed the discussion to point out that we do a rough estimate rather than a process based detailed analysis.

**While the sequestration cannot be said to be "shown", in my opinion, I do**

**think the authors could safely say that their results 'indicate a potential' CO2-sequestration of X t C/ha/yr.**

We concur and toned down the wording a little bit here and there.

**3. A thorough analysis of the water balance for the mesocosms needs to be presented. The water balance needs to include an evaluation of the transport time for water through the mesocosms.**

The water balance data we have is shown in Fig. 2 of the MS. The transport time of water through the mesocosm was not tracked. We don't have any other information.

**4a. The overall purpose of the study appears to be slightly blurred. The focus on estimating CO2-sequestration rates infers that this was the main aim. (However, one must then ask why DIC in the effluent was not more carefully measured, ie., what were the mesocosms designed for?)**

This is a very good point, which we didn't address originally. The experiment was designed to evaluate elemental cycles in typical crops affected by rock powder application. The idea to evaluate weathering fluxes came in later. Therefore, the experiment seems not to be designed towards the questions raised in the manuscript. Also, this explains why DIC data is patchy: We could only use what was left over. To justify the "purpose-deviating" setup, we added a sentence in the beginning of the methods section.

**4b. Another purpose appears to be to demonstrate the use of dunite as a "model mineral" for enhanced weathering experiments. (But then, why the strong focus on the estimation of CO2-sequestration rates and trace elements, which implies a focus on field applicability?)**

We use the mineral because of its relative simplicity in terms of geochemical composition and thermodynamic response. As the complexity of weathering effects, resulting from deployment of large amounts of small grain size powder, is not understood at all

at the moment, we abstain for the moment from a more complex mixture of elements and minerals like in basalt or other more differentiated rock types. Yes, it is unlikely that dunite is used, if EW is deployed, but it can be a good starting point to identify the various pitfalls of field deployment. The overarching aim of the manuscript is to work out some general statements of what must be considered in future experiments, let alone open field deployment.

**4c. The authors should state the objective(s) more clearly.**

We sharpened the text, especially the last paragraphs of the introduction to be more precise.

**5a. Presentation and structure: The manuscript needs to undergo a major revision in terms of the structure, conciseness of the text, and its figures. For example, the manuscript contains many repetitions and many imprecise statements. Also, many results are presented in the Discussion and some discussion take place in the Results section. Some results were not presented (or did I miss them during my reading?) but were still referred to/used in the Discussion.**

We took your advice and reworked the text.

**5b. The artwork needs to be polished; generally, the figures in the supplementary information seem to be better worked through than the figures in the manuscript, although the depths (e.g., cm below soil surface) needs to be added in the supplementary material, rather than using a 'depth number'.**

We tried to streamline the artwork a bit. However, due to the complexity of the data, it is hard to find a good way of presenting it. We are open for further suggestions. The issue of depth indications was fixed in all figures and also in the text.

**5c. Five replicates for each treatment combination were conducted, but this need to be visualized by statistics in the figures.**

In general, we agree. However, due to the complexity of the data, we think the absence of error bars in the main text can be justified. It seems unreasonable to put the appendix figures into the main document, since it would expand the document significantly. Exemplary, we created the figure for Mg concentrations as in the manuscript this time including error bars (Rebuttal Fig. 1). As can be seen, the addition of error bars decreases readability, but does not influence the major conclusions from the mean values presented in the figure in the main text.

*Rebuttal Fig. 1: Recreation of Fig. 5 of the main manuscript to show readability reduction through inclusion of error bars. Green: fine olivine, red: coarse olivine, blue no olivine.*

*Comments within the manuscript*

**Please use lower case: "enhanced weathering", to comply with the most frequent use in the literature.**

We explicitly chose to write it capitalized to indicate that we don't talk about the process but a "method" of carbon sequestration. In previous studies this style of writing was already adapted, to show the difference between the geological process and the method.

**The authors should comment on the differences between "unplanted+olivine" and "unplanted-no olivine". Could indicate significantly different storage properties. If so, it is not a total game changer, but still needs to be understood/explained.**

It looks truly interesting if we leave away the indicators for standard deviations. Yet, the variance is very high and a statistical test (Mann Whitney U) reveals that differences are rarely significant at the 5

*Rebuttal Fig. 2 Exemplary comparison of statistical differences between outflow volumes of mesocosms with fine, coarse and no olivine treatment. Mann-Whitney-U p-Values below the red line are <0.05 and therefore considered to indicate a significant*

*outflow volume difference in treatments.*

**This** [preferential flow effect; author's note ] **is an important point in the manuscript. The underlying reasoning (for factoring up the inferred CO2-sequestration) is that lots of weathering take place in the still standing solute which is claimed to be continuously bypassed by the macropore flow. However, the net weathering would still be expressed by the effluent water flux multiplied by its weathering product concentration and a stoichiometric coefficient. The deduction of a preferential flow factor therefore is, as far as I am convinced, useless for the purpose of estimating the CO2-sequestration. [. . .]. Preferential flow may play a role, but will not change the final weathering rate. DIC needs to percolate to the groundwater table and beyond, and/or form (stable) carbonate minerals within the soil. The latter is not shown.**

This is an important point. The final weathering rate is of course determined by what leaves the system. This is why we provide the "observed" sequestration rate. Also, we don't specifically mention the higher weathering rate, factoring out the preferential flow effect, in the main text of the discussion, to avoid inconsiderate use of the number. However, we think that the experiment points out nicely, that there are probably processes in play, which were not considered so far, and we suggest that preferential flow is one of them. To point out the differences more clearly, we changed the wording, to be more precise and to avoid misunderstandings in the direction of calculated sequestration rates. Since we changed a lot of words all over the manuscript, it is not possible to point out specific sections.

**What the authors state [in the hydrology section; author's note] is that there was no relation between irrigation scheme and weathering rate, CO2 uptake, etc. (all other monitored parameters). This is an important result! It should be emphasised more and data or statistics should be presented for support.**

We created an exemplary figure to show the differences resulting from the two rain

regimes and added a remark in section 3.1 and supplied the comparison in the supplementary material (Suppl. Fig. S2-1). It can be deduced from the Mg data (selected as example), that there are some points where differences between rain treatments are significant but generally, they are not. Furthermore, there is no systematic deviation between the rain treatments. This is why we decided to lump them together in the study at hand.

―――――――――――――――――

[Figure]

**Fig. 1.**

[Figure]

**Fig. 2.**

[Figure]

---

## Author Comment (AC3) · 29 Mar 2019

Dear David, we are grateful for your thorough review of our manuscript. In the following we document the changes that we made according to your suggestions.

**Reviewers comment**

Our reply

**1) 22kg/m2 (page 4) is a very high application rate! The plants will be growing in olivine. Is this the correct application rate or has a gremlin affected the units?**

The number is correct. A high application rate was chosen for this experiment, to

induce strong and quickly observable effects. The value is similar to the maximum amount applied in a pot experiment by ten Berge et al. (2012). We added an explanatory sentence, to make sure no one stumbles upon this value. It is safe to say the Gremlins behaved well here.

**2) I note you state the source of the olivine. Are there any references to other pub- lished descriptions of the material in the ÅĿheim deposit? It would be good if these could be provided, as you only give basic information concerning the mineralogy and geochemistry.**

We adapted the text to a more scientific term for the origin of the sample (Almklovdalen peridotite complex) and added a few references discussing the geological and geochemical background of the material (L 2-10ff)

**3) With that in mind, a good reference to the deposit might address these queries concerning Table 1: a) why is the LOI so high?; b) could you recalculate the mineral composition that the chemical analysis represents?; c) is any asbestos associated with this material?; d) total iron is given as Fe2O3 (this should be stated), yet olivine contains divalent iron. What is the iron mineral in this material?**

Ad a) We assume that the comparably high LOI can be explained by the abundance of hydroxide bearing minerals like lizardite, chlorite, and amphibole, as well as hydration water bearing chabazite. A study of serpentinites of an ophiolite complex further north shows even higher LOIs of around 11Ad b) It would be possible to calculate the mineral composition based on certain norms like CIPW, however, due to the transitional characteristic of the material, a metamorphically altered magmatite, it seems unlikely that results will be interpretable. The standard CIPW norm does not consider the water content and will as such not deliver the observed minerals, listed in a). We therefore believe that our observations from the XRD analyses are more reliable. Ad c) We identified lizardite by XRD analyses. Lizardite falls into the group of asbestos minerals. Ad

d) As the XRF analysis conventionally reports the Fe content as Fe2O3, the exact split between Fe2+ and Fe3+ is unknown. Most of the iron is probably divalent, from olivine and chlorites.

**4) Some typos: p4 line 15 - X-ray not x-ray; p5 line 10: magnesium not Magnesium; p6 line 21: through not trough. Check once more for other typos elsewhere!**

We took care of this and checked the text once more.

**5) Back to the science: do you have Si and Mg data for the plant biomass? I think this is important to give a mass balance of removal of these elements from the soil and its constituent minerals.** The original study for which the experimental setup was designed deals with the effect of the olivine application on plant growth and productivity. Results are processed in another MS that is currently being finalized. We can therefore not include them in the present MS. Moreover, the present MS focusses on the weathering rates and potential for inorganic carbon sequestration and adding information about the plants would distract from the main focus. It will also elaborate the paper considerably (more material and methods, more figures, more discussion points, etc.) which is not desirable and is the main reason for us to divide the results in 2 publications.

**6) Did you find any evidence of precipitation of Mg carbonate minerals, as reported for 'similar' rocks by Dipple's group? Did you look for these minerals?**

As we looked only into the aquatic phase, we have no evidence of Mg mineral precipitation. During the summer the mesocosms upper parts dried out fully. Therefore, temporary precipitation is very likely. As we explained in the response to reviewer 1, the experiment was originally designed as a plant experiment, and no soil samples were taken during the summer period.

**7) Both Cr and Ni are essential nutrients for a range of biological processes. I'd**

**prefer to avoid the use of the emotive word 'contaminant' as we'd all be dead without sufficient of these elements in our diet.**

We fully agree. This is why we avoided the word. We were not able to find any occurrence in the manuscript.

**8) Again, do you have any evidence for differential uptake of these elements into the crops? If there is no significant difference between treatment and control, then you have no evidence of a problem.**

In the manuscript, we only discuss the release and mobility of Cr and Ni in the water phase. ICP analysis shows indeed, that concentrations of Cr or Ni in any analysed plant part (stem, leave, grain) is below the detection limit of about 2 ppm.

**9) What was the mineralogical composition of the soil that was used? This should be stated, to ensure that any confounding factors (such as preferential weathering of a soil mineral already there) can be assessed. I appreciate the design of the study would avoid such factors, but it would be very useful to know. For example, does the soil contain carbonate minerals?**

All data available to us was added to Suppl. Mat. S1 to generate a more detailed characterisation of the used soil.

*References*

Beyer, E. E.: Transformation of Archaean Lithospheric Mantle by Refertilization: Evidence from Exposed Peridotites in the Western Gneiss Region, Norway, Journal of Petrology, 47, 1611-1636, 10.1093/petrology/egl022, 2006.

Iyer, K., Austrheim, H., John, T., and Jamtveit, B.: Serpentinization of the oceanic lithosphere and some geochemical consequences: Constraints from the Leka Ophiolite Complex, Norway, Chemical Geology, 249, 66-90, 10.1016/j.chemgeo.2007.12.005, 2008.

ten Berge, H. F. M., van der Meer, H. G., Steenhuizen, J. W., Goedhart, P. W., Knops, P., and Verhagen, J.: Olivine Weathering in Soil, and Its Effects on Growth and Nutrient Uptake in Ryegrass Lolium perenne L.: A Pot Experiment, PLoS ONE, 7, e42098, 10.1371/journal.pone.0042098, 2012.

---

## Author Response (AR3)

**Author's response**

We like to thank reviewer Søren Jessen for his thoughtful second review. We acknowledge the issues brought up and, in the following, we reply to the points raised.

We understand that the reviewer's primary concern is our assumption that basically outflow Mg concentrations are only low because of dilution with water that was not or only shortly in contact with the soil due to preferential flow paths.

We acknowledge the critique, and agree we cannot present evidence that this is the only process responsible for 13-fold lower Mg concentrations in outlet water when compared to the olivine-treated surface layer. We therefore now elaborate on different processes that can lead to these differences, via dilution, removal, or up-concentration in the surface. We included 1-2 brief sentences to every possible process at play, while keeping it qualitative as our dataset does not allow the quantification of responsible processes. The changes are primarily to be found in section 4.2, second, third, and fourth paragraph. This of course has slightly changed the overall conclusions of our research. Therefore, modification to the abstract and concluding paragraph were made, which involve mainly toning down some of our claims.

Also, circular and overly speculative conclusions were mentioned. We screened the text, especially the regarding the issue raised above and included more potentially involved processes to balance the previously less substantiated, speculative unidirectional statements.

A few specific points were mentioned, which we reply to here. Further small changes were made throughout the manuscript which can be best checked in the document with tracked changes, attached at the end.

**Reviewers comment**
*Our reply*

**If I am not mistaken, your analysis uses the concentration differences ("in"-"out") of the non-conservative Mg to calculate a dilution/preferential flow effect, which is then used to calculate CO2 uptake. (If I am mistaken, please change the manuscript so that others don't get the same impression is I do.)**
**This approach warrants a much more careful treatment and discussion, than present in the current version of the manuscript.**

You were not mistaken. There is a strong Mg difference between surface and outlet. We discuss this a bit broader now, naming preferential flow only as one possible process responsible.

**p 16, l 26-28: Please explain exactly why "it is unlikely"**

This was removed as we cannot explain it.

**and also include "adsorption" in your listing in the end of the sentence.**

We included the process in the discussion.

**Also, very low volumes of water actually comes through the mesocosms (or am I mistaken here too?), so the phrase 'large quantities of water bypassing ...' should be exchanged for something else.**

This part was removed as well, as the discussion is not primarily focusing on preferential flow any longer.

**This brings me to an alternative explanation for the Mg difference: The high Mg concentrations at 1.5 cm depth could be due to evaporation as the removal of water may leave Mg in solution at elevated concentrations (cf. Thaysen et al., 2014a (doi.org/10.5194/bg-11-7179-2014) and 2014b (doi.org/10.2136/vzj2014.07.0083)). Very low water volumes actually passes through the column despite continuous irrigation, and no water balance which can account for the residence time was or could be made. Who knows, maybe water moves upwards in large parts of the column? Therefore, please elaborately discuss this alternative explanation for the Mg concentration differences, and soften your statements accordingly throughout the manuscript.**

You are right and evapotranspiration is surely a process that can be expected. We included this and added a reference to one Thaysen paper also describing the effect.

**For example, the first time the dilution is mentioned is in bi-sentence (..., recognizing processes that lead to dilution, ...; p 13 l 6-7) by which time the reader apparently has to just accept this as a fact?**
**I probably just have overlooked a point in your work - if so, please argue fully and carefully in the text of the diminutive effects of the non-conservative Mg behavior and evaporation.**

As described in the beginning, we toned everything down and added more processes to the discussion.

**Several of the authors are highly experienced researchers, who I am sure are able to help avoid 'circular conclusions' and 'overly speculative conclusions' in the final version. (Excuse my direct address here, please.)**
**An example of what I see as a circular conclusion appears in the abstract (p 1 l 21-23): "Calculations that explicitly factor out the dilution effect of bypassed water, lead to relative fluxes about a magnitude higher, confirming that preferential flow paths and surface runoff in the field must be included in assessments for the CO2 consumption potential of Enhanced Weathering in general." The statement appears to assume that the previous CO2 sequestration rate is correct and that a rate ten times higher cannot be correct?**
**An example of what I see as an 'overly speculative conclusion' appears also in the abstract: Here I agree and find it correct to conclude that "Porewater Mg/Si molar ratios suggest that dissolved Si from the added minerals stays in the system over the observation period,..." but the reason that follows is not warranted: "...because a cation depleted Si layer forms on the reactive mineral**

**surface of freshly ground rocks.". And the anticipation of such a layer does not prove its actual existence, which is strongly inferred in the sentence that then follows: "This layer has not reached equilibrium thickness within the first two years." For a reader of the abstract it must be made clear that no direct observation of the Si-enriched layer was observed, but only inferred from water chemistry. Again, perhaps I missed a point. If so, when you correct the text, please state the method by which layer was observed.**

You are right and the text was screened for speculativeness and circular conclusions. We hope, we toned it down in a way that it can satisfy the readership. The specific text parts mentioned above were changed in order to reflect that we do not have any direct observations.

[revised manuscript text omitted]

---

## Author Response (AR4)

**Author's response**

Dear Editor,

Thank you for still considering our manuscript. In the following, we will document the requested minor changes.

**Reviewers comment**
*Our reply*

**This change should be reflected in the title that promises constraints on enhanced weathering, which were, given the uncertainties, not as tight as one might have wished.**

*We agree and changed the title to be more general.*

**You write "Enhanced Weathering" throughout the text. Please don't use upper case letters, maybe rather use and abbreviation (not in the abstract) or a symbol.**

*We defined the term once in the abstract and main text and use the abbreviation thereafter.*

*We understand that it is common not to capitalize the first letters. However, as this study is on a rather novel approach in the climate change community, we like to establish a capitalized writing of „Enhanced Weathering". We had this issue already on the context of another publication in Biogeosciences (https://www.biogeosciences.net/16/2949/2019/) and the capitalized version was accepted there after some discussion with the copy editing compartment.*

*We are aiming to act on former discussions with colleagues from the field to create something similar to a capitonym here. There is a strong difference in the meaning between capitalized and non-capitalized reproduction of the words. The non-capitalized version merely describes a rather generic weathering process, which could be of chemical or physical nature, and enhanced by all kinds of things. Yet, the capitalized version implies much more than just a weathering process - it means a chemical weathering process embedded in a very specific techno-economically constrained method of CO2 removal. Therefore, the strong differences in meaning, much like Earth and earth, should justify the capitalization. Even if this was not or only partially done in earlier publications.*

*When looking for literature around the technology of EW, one finds a plethora of papers referring to the basic „process" of enhanced weathering with all kinds of meanings. This makes it cumbersome at times to identify the suitable references. A differentiation in writing can make all the difference. We understand that the journal tries to stick to basic orthographic rules and this is certainly a very good thing. But we like to appeal to the journals ambition to go one step further and beyond the basic science, to establish a new method here, with a proper and distinguishable name and way of writing.*

**(1|23) "one order of magnitude"**

*Changed.*

**(16|8) insert "were" before "around"**

*At the given position, this word couldn't be found. At position (8|16), the word is found, but "were" before it would not make sense. The sentence seems to be correct.*

**(16|25 and 17|28) is "downward percolation" not a tautology in a gravitational field? Consider omitting "downward" or explain, why that is a necessary specification.**

*Agreed and removed the "downward".*

**(19|26) replace "a magnitude" by "one order of magnitude"**

*Corrected here and at one more occasion (15|7).*

[revised manuscript text omitted]